# A Panchromatic Cyclometalated Iridium Dye Based on 2-Thienyl-Perimidine

**DOI:** 10.3390/molecules27103201

**Published:** 2022-05-17

**Authors:** Paulina Kalle, Marina A. Kiseleva, Sergei V. Tatarin, Daniil E. Smirnov, Alexander Y. Zakharov, Viktor V. Emets, Andrei V. Churakov, Stanislav I. Bezzubov

**Affiliations:** 1N.S. Kurnakov Institute of General and Inorganic Chemistry, Russian Academy of Sciences, Leninskii pr. 31, 119991 Moscow, Russia; kalle@igic.ras.ru (P.K.); marina.kiseleva@chemistry.msu.ru (M.A.K.); sergei.tatarin@chemistry.msu.ru (S.V.T.); daniil.smirnov@chemistry.msu.ru (D.E.S.); alexan.zakharov@yandex.ru (A.Y.Z.); 2Department of Chemistry, Lomonosov Moscow State University, Lenin’s Hills 1, 119991 Moscow, Russia; 3Frumkin Institute of Physical Chemistry and Electrochemistry, Russian Academy of Sciences, Leninskii pr. 31, 119071 Moscow, Russia; viktoremets@mail.ru

**Keywords:** iridium, cyclometalated complexes, perimidines

## Abstract

Though 2-arylperimidines have never been used in iridium(III) chemistry, the present study on structural, electronic and optical properties of *N*-unsubstituted and *N*-methylated 2-(2-thienyl)perimidines, supported by DFT/TDDFT calculations, has shown that these ligands are promising candidates for construction of light-harvesting iridium(III) complexes. In contrast to *N*-H perimidine, the *N*-methylated ligand gave the expected cyclometalated μ-chloro-bridged iridium(III) dimer which was readily converted to a cationic heteroleptic complex with 4,4′-dicarboxy-2,2′-bipyridine. The resulting iridium(III) dye exhibited panchromatic absorption up to 1000 nm and was tested in a dye-sensitized solar cell.

## 1. Introduction

Cyclometalated iridium(III) complexes have proved to be the best emitters in phosphorescent organic light-emitting diodes (PHOLED) due to their bright luminescence, appropriate excited state lifetimes, high thermodynamic and kinetic stability, and fine tuneability of the emission color by simple ligand variation [1,2]. These complexes are also considered as effective photocatalysts [3,4] and promising photosensitizers in solar cells [5,6,7,8,9,10,11,12,13,14]. Still, for the latter application, the light-harvesting properties of iridium(III) complexes need to be improved by judicious selection of cyclometalating and/or ancillary ligands.

Though first examples were reported more than 100 years ago [15], perimidines have been studied in detail only since the 1970s, when their unusual chemical behavior was explained in terms of their electronic structure [16]. Perimidines are fused nitrogen heterocyclic compounds in which the expulsion of a formally excessive π-electron from the six-membered heteroring towards the naphthalene fragment results in noticeable polarization of the π-electron cloud (Figure 1). So, the positive charge is mainly localized at the C2 carbon atom, while the naphthalene system hosts the negative charge. The charge transfer between these fragments of the molecule induces intensive low-energy absorption (ε ~ 10^3^ M^−1^cm^−1^) in the visible spectral range and determines bright yellow color of perimidines. The ability of perimidines to change drastically their color upon the variation of electron-donating/-withdrawing substituents in the position 2 [17] in combination with their strong π-electron donating properties [18] make them suitable “antenna” ligands for the design of stable light-harvesting cyclometalated iridium(III) complexes for application in solar cells and photocatalysis. Surprisingly, only few perimidines served as ligands in coordination compounds [19,20,21,22], among which there were no examples of C^N-cyclometalated complexes [23].

Herein, we synthesized and thoroughly characterized two 2-(2-thienyl)-perimidines containing thiophene rings, responsible for giving the ligands effective chromophore properties [24]. We found that the metalation of only the *N*-methylated ligand by iridium trichloride hydrate afforded the expected μ-chloro-bridged iridium(III) dimer in reasonable yield. X-ray study of the latter revealed significant bending of the cyclometalated ligands and strong intramolecular π-π stacking between them. The dimer was readily converted into a heteroleptic complex with 4,4′-dicarboxy-2,2′-bipyridine as the “anchoring” ligand. This novel iridium(III) dye demonstrated panchromatic absorption up to 1000 nm and was tested in a dye-sensitized solar cell.

## 2. Results and Discussion

### 2.1. Synthesis and Crystal Structures of Ligands

2-(2-Thienyl)-perimidine **L1** was prepared in high yield via condensation of 1,8-diaminonaphthalene with 2-thiophenecarboxaldehyde in the presence of sodium metabisulfite (Figure 1) [25,26]. *N*-methylation of **L1** was conducted with methyl iodide in alkaline media under inert atmosphere to prevent oxidation of the perimidine system (73% yield). All the compounds were characterized by ^1^H and ^13^C{^1^H} NMR (Appendix A), and single-crystal X-ray diffraction.

The asymmetric unit of the **L1** crystal consists of the main molecule and one solvent dichloromethane while that of the **L2** contains only the perimidine. In both structures, the main molecules comprise the flat perimidine system and the 2-thienyl ring which are twisted in a greater or lesser extent depending on the *N*-substituent and likely the packing effects (Figure 2, Appendix A). **L1** is almost flat while **L2** shows notably larger interplanar angle between the aromatic parts because of steric repulsion between the *N*-methyl group and the 2-thienyl ring. The hydrogen atom at the C13 atom imposes an additional strain in the molecule and makes the perimidine system and the *N*–CH_3_ non-coplanar bond (see Appendix A for details).

In the crystal of **L1**, molecules form centrosymmetric dimers through strong π···π stacking between the perimidine systems. These dimers are combined in layers lying in the *bc* plane by C–H···π interactions and moderate N–H···N hydrogen bonds. The resulting layers alternate with dichloromethane layers forming the 3D packing of the crystal (Appendix A). Similarly, the structure of **L2** contains π···π-bonded dimers grafted together in offset stacks by several C–H···π interactions, while weak π···π contacts between the thienyl rings combine the stacks in the 3D packing of the crystal (Appendix A).

### 2.2. Optical Properties of Ligands

Both ligands **L1** and **L2** are orange solids having similar UV-vis absorption spectra in acetonitrile (Figure 3). Large differences in the range from 250 to 320 nm and slight hypsochromic shift at longer wavelengths are very likely caused by the decrease in conjugation between the perimidine and 2-thienyl rings upon the introduction of the *N*-methyl substituent. So, the corresponding (380–500 nm) absorption bands can be attributed to the charge transfer between the above parts of the ligands.

### 2.3. DFT Calculations of Perimidine Ligands

Analysis of the optimized geometry of the ligands shows that even the unsubstituted **L1** is nonplanar (deviation from planarity reaches 8°), while **L2** is noticeably twisted along the C4–C5 bond, though the absolute values of the dihedral angles between the aromatic parts of the molecules are significantly smaller than those in the crystal structures (<34°). In contrast, the angle between the *N*–CH_3_ bond and the perimidine system in the optimized structure of **L2** (16°) exceeds the same angle obtained in the X-ray diffraction experiment (5°).

The highest occupied molecular orbitals (HOMO) of both perimidines are localized at the perimidine moiety exclusively, whereas the lowest unoccupied molecular orbitals (LUMO) are mainly delocalized over the two adjacent heterocycles with approximately equal contributions regardless of the *N*-substituent (Figure 4).

TDDFT calculations being consistent with experimental spectra give insights into the origin of absorption bands of **L1** and **L2** (Appendix A). The absorption of the perimidines in the visible spectral range is caused by the charge transfer from HOMO to LUMO, whereas strong bands observed at ca. 350 nm correspond to the π → π* electronic transitions localized at the naphthalene moiety, with the oscillator strength for all these bands being practically invariant to the geometry of the ligands. In contrast, the oscillator strength of the adjacent high-energy absorption band at ca. 290 nm corresponding to the charge transfer from the 2-thienyl ring to the heterocyclic unit of the perimidine drops dramatically in going from **L1** to **L2** that is the consequence of the decrease in conjugation between the heterocycles in **L2**.

### 2.4. Geometry and Electronic Structure of Iridium(III) Cyclometalated Chlorides Based on Perimidines: A DFT Study

Though the vast majority of C^N-cyclometalated iridium(III) chlorides are μ-chloro-bridged dimers [27], it is well documented that the increase in the steric pressure (for example, induced by phenanthroimidazole-based cyclometalated ligands) destabilizes the iridium octahedron and rare monomeric trigonal–bipyramidal chlorides can be isolated [28,29,30,31,32,33]. Given that the perimidine core is sterically more bulky than nitrogen-containing parts of cyclometalated ligands commonly used in iridium(III) chemistry (in most cases, pyridine ring in 2-phenylpyridines), and is rather similar to the phenanthroimidazole unit, the formation of the above monomeric species becomes possible.

In the case of the dimer based on perimidine **L2**, DFT calculations reveal distortions of the iridium octahedrons in combination with significant deformations of the cyclometalated ligands (Figure 5). The Ir–C distances (1.98 Å) lie within the expected range, whereas the Ir–N (2.13 Å) and Ir–Cl (2.64 Å) bonds are slightly longer than those in reported μ-chloro-bridged dimers (1.93–2.05, 1.99–2.11 and 2.43–2.61 Å, respectively) according to the Cambridge Structural Database (CSD, ver. 5.4, 3 November 2021). While the arrangement of the ligands is strictly controlled by the iridium(III) ion coordination preference the conformation of the ligands is additionally regulated by π···π stacking between the perimidine moieties. So, in the resulting conformation, the dihedral angle between the metalated thienyl ring and the perimidine system reaches 30°. In contrast, the optimized geometry of the pentacoordinated monomer with chloride lying in the equatorial plane contains almost planar aromatic ligands with deviations from planarity not exceeding 10°. Nevertheless, the dimer appears more stable than the monomer with the energy gain of 10 kcal·mol^−1^, indicating that the strong preference of the iridium(III) ion for octahedral geometry has a dominant effect on the structure of the cyclometalated chloride.

Frontier molecular orbitals of both the dimer and monomer are mainly localized at the ligands. The perimidine system hosts 70–80% of the HOMO and 40–50% of the LUMO (at the nitrogen heterocycles), while π-orbitals of the thienyl rings contribute mainly to the LUMO (~50%) and make lesser contributions to the HOMO (~10%). So, cyclometalation of the **L2** by the iridium ion, regardless of the product formed, does not result in noticeable redistribution of the electron density at either the ground or excited states of the cyclometalated ligand compared with its uncomplexed form.

### 2.5. Synthesis and Structure of Cyclometalated Iridium(III) Complexes with Perimidine Ligands

With the ligands and theoretical results in hand, we next aimed to synthesize the first examples of cyclometalated iridium(III) complexes incorporating 2-arylperimidines. Reaction of iridium(III) chloride hydrate with perimidine **L1** under standard Nonoyama conditions [34,35] led to precipitation of a large amount of iridium black, whereas the filtrate contained the unreacted ligand and a complex mixture which could scarcely be characterized. The change of the 2-ethoxyethanol/water mixture to pure 2-ethoxyethanol or variation of reaction temperature in the range 100–135 °C did not furnish any detectable amount of cyclometalated products.

In contrast to the perimidine **L1** which seemed to be more susceptible to oxidative destruction by the iridium(III) ion, reaction of the perimidine **L2** with iridium(III) chloride hydrate under the standard conditions (Figure 2) was not accompanied by the formation of iridium black and afforded a dark-red solution as well as a brown precipitate. According to ^1^H NMR (CDCl_3_), the solution contained only the unreacted perimidine **L2** (Appendix A). The precipitate was partially soluble in hot toluene and the ^1^H NMR spectrum (DMSO-*d*_6_) of the solution was discernibly different from that of the free ligand (Appendix A). While the uncoordinated 2-thienyl ring exhibited three complex signals at 6.71, 7.63 and 7.77 ppm in the spectrum of **L2**, the spectrum of the toluene extract contained two sharp, clearly resolved doublets at 6.63 and 7.58 ppm (J = 5.0 Hz) with relative intensity ratio 1:1 which could be assigned to two protons of the metalated 2-thienyl ring. Although the other aromatic signals were either broadened or heavily overlapped, which hampered the correct assignment, a sharp singlet of relative intensity 3 at 3.94 ppm seemed to correspond to the *N*-CH_3_ group.

The toluene solution was characterized by high-resolution mass spectrometry and signals corresponding to the [Ir(C^N)_2_]^+^ and [Ir(C^N)_2_(CH_3_CN)]^+^ molecular ions were detected (Appendix A). These charged species could be formed by removal of chloride (or through the change of chloride to acetonitrile) not only from the μ-chloro-bridged dimer, but also from the pentacoordinated monomer.

Eventually, the X-ray analysis unambiguously confirmed the dimeric structure of **1** which was very similar to its optimized structure (Figure 5, Appendix A). The Ir–C bond lengths are slightly shortened (1.956(6)–1.968(6) Å) compared with the average value, according to the CSD, the Ir–N distances, on the contrary, are slightly elongated (2.088(5)–2.098(5) Å), while the Ir–Cl bond lengths lie within the standard range from 2.5109(14) to 2.5199(13) Å. The reason for these deviations is likely the result of simultaneous influence of several factors, including the strong preference of iridium(III) for the octahedron, steric strain in the cyclometalated perimidines exerted by the methyl groups (the angle between the N–CH_3_ bond and the perimidine plane is in the range 7.1(4)–16.9(5)°), and strong intramolecular π-stacking (d(π–π) = 3.436(7) and 3.234(10) Å) between the fused aromatic rings of C^N ligands coordinated to different iridium(III) ions. The combination of these factors also causes significant distortions of the geometry of the ligands so that the dihedral angle between the planes of the metalated 2-thienyl and perimidine reaches 33.0(5)°, that even exceeds the values in the optimized structure of the dimer.

The reaction of the dimer **1** with 4,4′-dicarboxy-2,2′-bipyridine proceeded efficiently to afford the desired heteroleptic complex **2** (Figure 2) which was characterized by ^1^H, ^13^C{^1^H} NMR (Appendix A) and HRMS.

In the mass spectrum, signals of the [Ir(C^N)_2_(H_2_dcbpy)]^+^ and [NH_4_][Ir(C^N)_2_(Hdcbpy)]^+^ molecular ions were detected (Appendix A). Addition of concentrated solution of NH_4_PF_6_ in methanol to the solution of **2** in the same solvent gave single crystals which were studied by X-ray crystallography.

The splitting of the dimer **1** followed by the formation of the complex **2** is concomitant with the disappearance of the intramolecular π-stacking resulting in relief of steric strain around the iridium(III) ion though the coordination geometry remains distorted octahedral (Figure 6). The Ir−C and Ir–N_perimidine_ bond lengths are almost the same as those in the dimer **1**, while two nitrogen atoms of H_2_dcbpy lying in *trans*-positions to the metalated carbon atoms are distant from the central ion at 2.1405(19) and 2.1628(19) Å. Bond angles about the iridium(III) ion vary within the ranges 81.57(7)–107.52(7) and 168.57(7)–173.35(8)° (Appendix A). In the crystal, complex cations directly interact with each other only by C–H···π contacts between the perimidine moieties. Bridging solvent methanol molecules hold complexes together in hydrogen-bonded chains passing along the *b*-axis, in which the cations alternate with hexafluorophosphate anions (Appendix A). The latter do not show rotational disorder because of moderate O–H···F bonds with solvent methanol molecules (*d*(O···F) = 2.904(4) Å). Pairs of the adjacent chains related to one another by the inversion center form the 3D packing of the crystal with wide channels along the *b*-axis filled by solvent dichloromethane molecules.

### 2.6. Optical and Redox Properties of Complex **2**

Heteroleptic complex **2** isolated as a completely black powder turns dark red in solution (acetonitrile) and exhibits a UV-vis spectrum which at least in the visible range is drastically different from anticipated for bis-cyclometalated iridium(III) complexes [27] (Figure 7). In addition to the common intraligand π → π* electronic transitions in the UV range and less intensive MLCT bands at 450–550 nm the complex surprisingly possesses several bands at 650–950 nm. Deconvolution of the spectrum into its Gaussian components reveals six individual bands in the visible range with ε varying from 200 to 2000 M^−1^cm^−1^ (Table 1). Though each band is of moderate intensity the cumulative absorption efficiency is high enough to consider this complex as a promising photosensitizer for dye-sensitized solar cells.

Complex **2** exhibits irreversible redox behavior in acetonitrile with one oxidation wave at +0.54 V and two reduction waves at −0.90 and −1.32 vs. Fc^+^/Fc (Appendix A). While the oxidation potential of **2** is high enough for the dye to be spontaneously reduced by the common triiodide/iodide redox mediator during the operating of DSSC the poor electrochemical reversibility of **2** is likely to dramatically decrease the device’s long-term stability.

A TiO_2_-coated photoanode sensitized by complex **2** is completely black and covers the whole visible range and even the near-infrared region, according to its diffuse reflectance spectrum (Appendix A). However, the photoanode achieves a photovoltage of only 0.13 V vs. NHE and a photocurrent of 0.12 mA/cm^2^ under simulated AM 1.5 G illumination, giving the negligibly small overall efficiency (Table 1, Appendix A). The incident photon to current conversion efficiency (IPCE) spectrum of the photoanode matches its diffuse reflectance spectrum and solution UV-vis spectrum of complex **2**, but the IPCE does not exceed 1% in the visible range (Appendix A). The poor overall device performance may be caused by insufficient absorption in the visible range and, more importantly, by the irreversible oxidation of **2**, which may be enhanced at the semiconductor surface [36].

### 2.7. DFT/TDDFT Calculations of Complex **2**

To gain more insights into the electronic structure of complex **2,** a joint DFT/TDDFT study was conducted. Analysis of the composition of the frontier molecular orbitals in the optimized structure of the complex shows that the HOMO and HOMO–1 (differing in energy by only 0.05 eV) are almost exclusively localized on the cyclometalated perimidines, while the LUMO is mainly localized on the “anchoring” 4,4′-dicarboxy-2,2′-bipyridine, and all these orbitals have very small contributions of iridium *d*-orbitals (Figure 8). The latter is in line with the experimentally observed irreversible oxidation of the complex because the redox process relates to electron removal from the orbital of essentially ligand character. It is interesting that the metal *d*-orbital contributions to the low-lying occupied molecular orbitals (from HOMO–2 to HOMO–5) do not exceed 27%.

TDDFT calculations show that absorption bands in the visible range, including the unexpected low-energy bands, originate from the electronic transitions from HOMO−1, HOMO−2 and HOMO−3 to LUMO and LUMO+1, and, hence, can be attributed to the interligand charge transfer from cyclometalated ligands to 4,4′-dicarboxy-2,2′-bipyridine (Appendix A, Appendix A). This charge transfer directionality likely favors effective electron injection from the dye to the TiO_2_ conduction band.

However, since the degree of overlap between π-orbitals of these ligands is minimal (because of their mutual practically orthogonal arrangement in the iridium octahedron), and while the metal *d*-orbitals obviously do not act as an effective π-bridge between these ligands, the probability of the corresponding electronic transitions is low, which reduces the intensity of the visible light absorption.

Still, the calculations provide valuable insights into the electronic structure of the complex **2** offering a possible way to enhance light-harvesting properties of similar iridium(III) complexes. Indeed, an increase in the metal *d*-orbitals contribution to the occupied molecular orbitals will apparently stimulate the interligand electronic transitions, cause the appearance of additional metal-to-ligand charge transfer bands in the UV-vis spectrum and may be concomitant with the achievement of reversible electrochemical behavior for the complexes. The desired redistribution of the HOMO electron density (as well as the other molecular orbitals participating in the low-energy transitions) may be triggered by the change of the 2-aryl unit in cyclometalating perimidines using a virtually unlimited set of aromatic aldehydes in the synthesis of 2-arylperimidines and/or by the variation of the “anchoring” ligand.

## 3. Materials and Methods

### 3.1. General Comment

All commercially available reagents were at least reagent grade and used without further purification. Solvents were distilled and dried according to standard procedures. 4,4′-Dicarboxy-2,2′-bipyridine was prepared as previously described [37]. Preparation of iridium(III) complexes was carried out under dry argon. Purification and other manipulations with complexes were performed in air.

^1^H and ^13^C NMR spectra were acquired at 25 °C on a Bruker Avance 400 instrument and chemical shifts were reported in ppm referenced to residual solvent signals. High-resolution, accurate mass measurements were carried out using a BrukermicroTOF-QTM APPI-TOF (Atmospheric Pressure PhotoIonization/Time of Flight) spectrometer. Electronic absorption spectra were measured on an OKB Spectr SF-2000 spectrophotometer. An Econix-Expert Ltd. Ecotest-VA polarograph was used for electrochemical measurements with a glassy carbon working electrode, platinum counter electrode, and saturated Ag/AgCl reference electrode. Polarographic curves were recorded in Ar-saturated acetonitrile with 0.1 M (*n*-Bu_4_N)ClO_4_ at a scan rate of 100 mV/s. Ferrocene was used as an internal standard.

### 3.2. Synthesis

2-(2-Thienyl)-1H-perimidine (**L1**). 1,8-Diaminonaphthalene (1.027 g, 0.0065 mol), 2-thiophenecarboxaldehyde (0.607 mL, 0.0065 mol) and Na_2_S_2_O_5_ (3.705 g, 0.0195 mol) in ethanol (30 mL) were refluxed under argon for 2 h. Orange solution was evaporated to dryness, extracted with methylene chloride (10 mL), filtrated to remove inorganic salts and the resulting solution was evaporated to dryness. Recrystallization from hot toluene gave 1.298 g of orange crystals (yield 80%).

^1^H (400 MHz; CDCl_3_; δ): 10.70 (s, 1H), 7.92 (d, J = 3.3 Hz, 1H), 7.76 (d, J = 5.0 Hz, 1H), 7.22 (m, 1H), 7.14 (t, J = 7.6 Hz, 2H), 7.02 (d, J = 8.1 Hz, 2H), 6.58 (br. s, 2H).

^13^C{^1^H} (101 MHz; CDCl_3_; δ): 148.3, 144.7, 138.4, 135.1, 130.8, 128.9, 128.2, 128.0, 127.7, 121.4, 119.3, 117.9, 113.7, 102.6.

Mp = 165–166 °C.

1-Methyl-2-(thienyl-2)-1H-perimidine (**L2**). To dry dimethylformamide (20 mL) **L1** (0.6 g, 0.0024 mol) and 60% sodium hydride (0.127 g, 0.0032 mol) were added, and the resulting slurry was stirred under argon for 10 min. Methyl iodide (0.15 mL, 0.0024 mol) was added and the dark-red mixture was stirred for 1 h. The solvent was removed under vacuum and the residue was purified by column chromatography (SiO_2_, CH_2_Cl_2_/hexane 1:3 → 1:1). Recrystallization from CH_2_Cl_2_/hexane (1/1) gave 0.460 g of orange crystals (yield 73%).

^1^H (400 MHz; CDCl_3_; δ): 7.47 (dd, J_1_ = 5.1 Hz, J_2_ = 1.0 Hz, 1H), 7.39 (dd, J_1_ = 3.7 Hz, J_2_ = 1.0 Hz, 1H), 7.32–7.28 (m, 1H), 7.24–7.17 (m, 3H), 7.12 (dd, J_1_ = 5.1 Hz, J_2_ = 3.7 Hz, 1H), 6.94 (dd, J_1_ = 7.3 Hz, J_2_ = 1.0 Hz, 1H), 6.30 (dd, J_1_ = 7.1 Hz, J_2_ = 1.2 Hz, 1H), 3.35 (s, 3H).

^13^C{^1^H} (101 MHz; CDCl_3_; δ): 150.8, 145.6, 140.2, 137.2, 134.6, 128.4, 127.8, 127.1, 126.6, 121.7, 119.9, 119.1, 115.1, 101.5, 37.0, 29.3.

Mp = 120–121 °C.

Di-(μ-chloro)-bis-(1-methyl-2-(2-thienyl)-perimidinato)diiridium(III) (**1**). **L2** (0.122 g, 0.46 mmol) and IrCl_3_·3H_2_O (0.074 g, 0.21 mmol) were mixed in 4 mL of 2-ethoxyethanol and heated at 110 °C under argon for 18 h. A brown precipitate formed was collected, washed several times with ethanol (3 mL) and ether (5 mL), and dried. The solid was extracted several times by a hot mixture of chloroform/toluene (3/1) and the combined red extracts were evaporated to dryness (yield 0.066 g, 42%). The product was practically insoluble in common NMR solvents, so it was used in the next step without characterization and any further purification.

HRMS (APPI) *m*/*z*: [M]^+^ calcd for C_32_H_22_N_4_S_2_Ir^+^ 719.0915; found 719.0911; [M+CH_3_CN]^+^ calcd for C_34_H_25_N_5_S_2_Ir^+^ 760.1181; found 760.1172.

Bis-(1-methyl-2-(2-thienyl)-perimidinato)-(4,4′-dicarboxy-2,2′-bipyridine) iridium(III) triflate (**2**). Compound **1** (0.060 g, 0.04 mmol) and 4,4′-dicarboxy-2,2′-bipyridine (0.020 g, 0.08 mmol) in a mixture of CH_2_Cl_2_ / MeOH 3/1 (20 mL) were refluxed under argon for 10h. Dark-red mixture was filtered and evaporated to dryness. The product was purified by column chromatography (SiO_2_, CH_2_Cl_2_ → EtOH/CF_3_SO_3_H 100:1) and isolated as triflate [Ir(**L2**)_2_(H_2_dcbpy)]^+^[CF_3_SO_3_]^−^. Yield 0.080 g, 90%.

^1^H (400 MHz; CD_3_OD; δ): 8.75 (m, 2H), 8.22 (d, J = 5.7 Hz, 2H), 8.17 (dd, J_1_ = 5.7 Hz, J_2_ = 1.5 Hz, 2H), 7.65 (d, J = 5.0 Hz, 2H), 7.27 (t, J = 8.0 Hz, 2H), 7.12 (d, J = 8.2 Hz, 2H), 7.03 (d, J = 5.0 Hz, 2H), 6.95 (d, J = 8.1 Hz, 2H), 6.64 (d, J = 7.7 Hz, 2H), 6.51 (t, J = 8.0 Hz, 2H), 5.82 (d, J = 7.7 Hz, 2H), 4.05 (s, 6H).

^13^C{^1^H} (101 MHz; CD_3_OD; δ): 164.8, 163.9, 157.6, 155.7, 148.9, 140.8, 140.6, 136.5, 134.3, 133.8, 133.3, 131.3, 127.5, 127.1, 126.8, 122.8, 120.8, 120.4, 119.5, 112.2, 104.7, 37.8.

HRMS (APPI) *m*/*z*: [M]^+^ calcd for C_44_H_30_N_6_O_4_S_2_Ir^+^ 963.1399; found 963.1389; [M−H+NH_4_]^+^ calcd for C_44_H_33_N_7_O_4_S_2_Ir^+^ 980.1665; found 980.1739.

Mp = 280 (dec.).

### 3.3. X-ray Crystallography

#### 3.3.1. Crystal Growth Conditions

**L1**: Slow evaporation of the solution of perimidine **L1** in dichloromethane.**L2**: Slow evaporation of the solution of perimidine **L2** in dichloromethane.**1**: Recrystallization of complex **1** from hot toluene.**2**: Addition of the saturated methanolic solution of NH_4_PF_6_ to the solution of complex **2** in dichloromethane/methanol mixture.

#### 3.3.2. Crystallography Details

Crystallographic data were collected on Bruker SMART APEX II (perimidines **L1** and **L2** at T = 100 K) and D8 Venture (complexes **1** and **2** at T = 100 K) diffractometers using graphite monochromatized Mo–Kα radiation (λ = 0.71073 Å) using a ω-scan mode. Absorption correction based on measurements of equivalent reflections was applied [38]. The structures were solved by direct methods and refined by full-matrix least-squares on F^2^ with anisotropic thermal parameters for all non-hydrogen atoms. In the structure of **L1**, hydrogen atoms were found from the difference Fourier map and refined freely. In the other structures, hydrogen atoms were placed in calculated positions and refined using a riding model. In the structure of **1**, two highly disordered solvent toluene molecules were not located and their contribution was suppressed by the SQUEEZE procedure [39]. In the structure of **L2**, the 2-thienyl ring was disordered over two positions with occupancies 0.55/0.45. Crystallographic details are presented in Appendix A and the crystal packings are plotted in Appendix A. CCDC 2032894, 2032893, 2169364 and 2169365 contain the supplementary crystallographic data for the structures **L1**, **L2**, **1** and **2**, respectively. These data can be obtained free of charge from The Cambridge Crystallographic Data Centre via www.ccdc.cam.ac.uk/data_request/cif.

Phase purity of the compounds was checked by X-ray powder diffraction recorded on a Bruker D8 Advance diffractometer (CuKα 1.5418 Å, Ni-filter, reflection geometry, 2Θ from 3 to 50°) equipped with a LynxEye silicon strip detector. Kα_2_ lines were removed, and the unit cell parameters were refined by least-square method using WinXpow program package. Theoretical patterns were calculated from the crystal data using Mercury software (ver. 3.1). Experimental and simulated powder diffraction patterns of **L2** are almost identical (Appendix A, Appendix A), whereas those of **L1** are significantly different (Appendix A). A possible reason is the removal, at least partial, of solvent dichloromethane molecules from **L1** that may result in change of its crystal packing. Solvent removal from **2** negatively affects its crystallinity because the solvent (methanol) participating in hydrogen bonds with main molecules plays crucial role in the crystal packing. So, the X-ray study revealed that the powder of **2** is substantially amorphous (Appendix A).

#### 3.3.3. Photovoltaic Measurements

Sensitization of titanium dioxide was performed by soaking of photoanodes (Solaronix) in ~5 × 10^−4^ M acetonitrile solutions of dye for 24 h. A three-electrode photoelectrochemical cell PECC-2 (Zahner) was used for the photoanode potential measurements. The photoanode served as the working electrode and a platinum wire with the surface area of 5 cm^2^ was used as the auxiliary electrode, a silver wire was used as the reference electrode. The voltammetric measurements were performed with an IPC Pro MF potentiostat under AM 1.5 global one sun of illumination (100 mW·cm^−2^) provided by a solar simulator (Newport 96000) in acetonitrile solution in the presence of 0.5 M LiI + 0.05 M I_2_. The illumination power at different distances was determined with a Nova apparatus (OPHIR-SPIRICON Inc., Ophir Optronics, Darmstadt, Germany). The current–voltage characteristic of DSSCs and the photocurrent density at the short-circuit voltage were performed by the two-electrode scheme. Transients of photoanode potential and photocurrent density at the short-circuit voltage were measured under irradiation and in dark condition. The photoanode area was 1.0 cm^2^. The illuminated photoanode area was restricted by a mask 0.196 cm^2^. The illumination was performed from the side of TiO_2_ photoanode with the adsorbed dye. Measurements of IMPS and IMVS as well as IPCE spectra were performed with a ZAHNER’s CIMPS-QE/IPCE workstation. The photoanode was illuminated with a tunable light source, TLS03. IMVS were taken without superposition of external polarization, i.e., under open circuit conditions. IMPS were recorded under short-circuit conditions.

#### 3.3.4. Density Functional Theory Calculations

All the gas-phase calculations reported in this paper have been performed within density functional theory (DFT) [40], using the hybrid functional B3LYP [41,42]. The standard def2-TZVP basis set for light elements and Stuttgart–Dresden effective core potential (ECP) for Ir atom, as implemented in the ORCA3.0 suite of programs [43], have been used together with the RIJCOSX approximation [44]. Frequency analysis was carried out to check whether optimized structures were local minima. No imaginary frequencies were found for local minima. Time-dependent DFT (TDDFT) calculations were carried out at the ground state geometries to obtain vertical excitation energies and theoretical absorption spectra. The lowest 30 singlet–singlet excitations were computed.

## 4. Conclusions

Based on analysis of the electronic structure of 2-arylperimidines, we predicted and then experimentally established that the involvement of perimidine-based cyclometalated ligands in the iridium(III) chemistry results in complexes having enhanced absorption in the visible spectral range. Although these ligands have never been used in construction of C^N cyclometalated metal complexes, we succeeded in synthesizing a μ-chloro-bridged dimer and a cationic heteroleptic iridium(III) complex with cyclometalated 2-(2-thienyl)-perimidine and “anchoring” 4,4′-dicarboxy-2,2′-bipyridine. The resulting dye demonstrated broad absorption covering the spectral range from the UV to the near-infrared region (up to 1000 nm). The excellent light-harvesting characteristics and, unfortunately, much poorer photovoltaic properties along with irreversible electrochemical behavior of the complex were explained through a joint effort of X-ray and photophysical experimental studies and DFT/TDDFT calculations.

The iridium-perimidine tandem seems promising for construction of efficient iridium dyes for application in dye-sensitized solar cells because the system is amendable to various structural and electronic modifications through the change of the 2-aryl unit in the ligand, which is available via a simple one-step synthetic procedure. Future work will focus on preparation of iridium(III) complexes possessing strong light-harvesting properties along with reversible redox behavior by rational design of 2-arylperimidines acting as cyclometalated and, possibly, ancillary ligands.

## Data Availability

Data is contained within the article and SI.

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
