# Peer review of "A Panchromatic Cyclometalated Iridium Dye Based on 2-Thienyl-Perimidine"

_molecules, 2022, doi:10.3390/molecules27103201_

Round 1

Reviewer 1 Report

Kalle et al present an interesting report on a panchromic cyclometalated iridium dye based on 2-thienyl-perimidine and its application in a DSSC. The report is well written and the study appears to be expertly carried out and coherently described. While a device with a disappointing IPCE was prepared, there are hints of potentially interesting chemistry here that should be of interest to the chemistry community. I recommend acceptance pending addressing a few minor points.
Most notably, what is the cause of the low IPCE? Can the authors speculate as to the reasons they suspect only low photocurrent is achieved? Is it simply due to the low absorptivity coefficient in the low energy regime?
Also, what is the rationalization for the red shift for the lowest energy front of L1 compared with L2? A decrease in conjugation?
 A few other minor points:
Line 147 – “kkal”
I was a bit confused as to why the DFT optimization of 1 precedes its synthesis. Line 193 states “Eventually, the X-ray analysis unambiguously confirmed the dimeric structure of 1 which was very similar to its optimized structure (Figure 5).” Include comparison of bond lengths/angles in SI?
Please add experimental temperatures to Scheme 2.
Reference abbreviations – please check – e.g., “European” in ref 24; Inorganica in ref 36; “(<scp>iii</Scp> )” in ref 33?

Author Response

Dear colleague,

Thank you very much for careful reading of the manuscript and for your comments. Our responses are presented below:

  1. Comment: Most notably, what is the cause of the low IPCE? Can the authors speculate as to the reasons they suspect only low photocurrent is achieved? Is it simply due to the low absorptivity coefficient in the low energy regime?

Reply: The low IPCE may be caused by insufficient absorption in the visible range and, more importantly, by the irreversible oxidation of complex 2 which may be enhanced at the semiconductor surface.

  1. Comment: Also, what is the rationalization for the red shift for the lowest energy front of L1 compared with L2? A decrease in conjugation?

Reply: The red shift at longer wavelengths is very likely caused by the decrease of conjugation between the perimidine and 2-thienyl rings upon the introduction of the N-methyl substituent.

  1. Comment: Line 147 – “kkal”

Reply: fixed

  1. Comment: I was a bit confused as to why the DFT optimization of 1 precedes its synthesis.

Reply: It was reported that the use of sterically demanding C^N ligands led to formation of pentacoordinated iridium compounds. We performed initial DFT optimization to estimate the geometry and energy for possible products (octahedral dimer / trigonal-bipyramidal monomer).

  1. Comment: Line 193 states “Eventually, the X-ray analysis unambiguously confirmed the dimeric structure of 1 which was very similar to its optimized structure (Figure 5).” Include comparison of bond lengths/angles in SI?

Reply: The bond lengths/angles of the calculated structure were added to the Table S2.

  1. Comment: Please add experimental temperatures to Scheme 2.

Reply: Experimental temperatures were added to the Scheme 2.

  1. Comment: Reference abbreviations – please check – e.g., “European” in ref 24; Inorganica in ref 36; “(<scp>iii</Scp> )” in ref 33?

Reply: References were changed according to the comments.

Reviewer 2 Report

In this contribution, the authors firstly exploited 2-arylperimidine scaffold in the design of cyclometallated Ir(III) complexes. It was found that the 2-arylperimidine ligands are promising candidates for construction of light-harvesting iridium(III) complexes. In contrast to the N-H perimidine, the N-methylated ligand gave the expected cyclometalated μ-chloro-bridged iridium(III) dimer which was readily converted to a cationic heteroleptic complex with 4,4′-dicarboxy-2,2′-bipyridine. The resulting iridium(III) dye exhibited panchromatic absorption up to 1000 nm and was tested in a dye-sensitized solar cell. Overall, the work seems to be sound, and the manuscript itself is well-written and illustrated. In my opinion, the reviewed work is high quality and of the proper impact/scope for this journal. Thus, I recommend this paper be published in this journal after minor revision to address one concern. The bulk purity of the synthesized CPs should be verified by powder X-ray diffraction analysis. What about photoluminescence of the complexes obtained? Are they emissive in solid or solution states?

Author Response

Dear colleague,

Thank you very much for careful reading of the manuscript and for your comments. Our responses are presented below:

  1. Comment: The bulk purity of the synthesized CPs should be verified by powder X-ray diffraction analysis.

Reply: Experimental and simulated powder diffraction patterns of L2 (not containing any solvent in the crystal structure) are almost identical (Figure S18, Table S4) whereas those of L1 are significantly different (Figure S19). A possible reason is the removal, at least partial, of solvent dichloromethane molecules from L1 that may result in change of its crystal packing. Solvent removal from complex 2 negatively affects its crystallinity because the solvent (methanol) participating in hydrogen bonds with main molecules plays crucial role in the crystal packing. So, the X-ray study reveals that the powder of 2 is substantially amorphous (Figure S20). The powder XRD patterns were added to the SI.

  1. Comment: What about photoluminescence of the complexes obtained? Are they emissive in solid or solution states?

Reply: The prepared perimidines and the corresponding iridium(III) complexes were not emissive both in solution (even in deaerated solvents) and in the solid state.